# Uncertainty Calibration Error: A New Metric for Multi-Class Classification

## Abstract

Various metrics have recently been proposed to measure uncertainty calibration of deep models for classification. However, these metrics either fail to capture miscalibration correctly or lack interpretability. We propose to use the normalized entropy as a measure of uncertainty and derive the *Uncertainty Calibration Error* (UCE), a comprehensible calibration metric for multi-class classification. In our experiments, we focus on uncertainty from variational Bayesian inference methods and compare UCE to established calibration errors on the task of multi-class image classification. UCE avoids several pathologies of other metrics, but does not sacrifice interpretability. It can be used for regularization to improve calibration during training without penalizing predictions with justified high confidence.

## 1 Introduction

Advances in deep learning have led to superior accuracy in classification tasks, making deep learning classifiers an attractive choice for safety-critical applications like autonomous driving (Chen et al., 2015) or computer-aided diagnosis (Esteva et al., 2017). However, the high accuracy of recent deep learning models alone is not sufficient for such applications. In cases where serious decisions are made upon model's predictions, it is essential to also consider the uncertainty of these predictions. We need to know if a prediction is likely to be incorrect or if invalid input data is presented to a deep model, e.g. data that is far away from the training domain or obtained from a defective sensor. The consequences of a false decision based on an uncertain prediction can be fatal.

A natural expectation is that the certainty of a prediction should be directly correlated with the quality of the prediction. In other words, predictions with high certainty are more likely to be accurate than uncertain predictions, which are more likely to be incorrect. A common misconception is the assumption that the estimated softmax likelihood can be directly used as a confidence measure for the predicted class. This expectation is dangerous in the context of critical decision-making. The estimated likelihood of models trained by minimizing the negative log-likelihood (i.e. cross entropy) is highly overconfident; that is, the estimated likelihood is considerably higher than the observed frequency of accurate predictions with that likelihood (Guo et al., 2017).

## 2 Uncertainty Estimation

In this work, we focus on uncertainty from approximately Bayesian methods. We assume a general multi-class classification task with $C$ classes. Let input $\boldsymbol{x} \in \mathcal{X}$ be a random variable with corresponding label $y \in \mathcal{Y} = \{1, \dots, C\}$. Let $\boldsymbol{f_w}(\boldsymbol{x})$ be the output (logits) of a neural network with weight matrices $\boldsymbol{w}$, and with model likelihood $p(y\!=\!c \,|\, \boldsymbol{f_w}(\boldsymbol{x}))$ for class $c$, which is sampled from a probability vector $\boldsymbol{p} = \boldsymbol{\sigma}_{\mathrm{SM}}(\boldsymbol{f_w}(\boldsymbol{x}))$, obtained by passing the model output through the softmax function $\boldsymbol{\sigma}_{\mathrm{SM}}(\cdot)$. From a frequentist perspective, the softmax likelihood is often interpreted as *confidence* of prediction. Throughout this paper, we follow this definition.

The frequentist approach assumes a single best point estimate of the parameters (or weights) of a neural network. In frequentist inference, the weights of a deep model are obtained by maximum likelihood estimation (Bishop, 2006), and the normalized output likelihood for an unseen test input does not consider uncertainty in the weights (Kendall & Gal, 2017). Weight uncertainty (also referred to as model or epistemic uncertainty) is a considerable source of predictive uncertainty for models

trained on data sets of limited size (Bishop, 2006; Kendall & Gal, 2017). Bayesian neural networks and recent advances in their approximation provide valuable mathematical tools for quantification of model uncertainty (Gal & Ghahramani, 2016; Kingma & Welling, 2014). Instead of assuming the existence of a single best parameter set, we place distributions over the parameters and want to consider all possible parameter configurations, weighted by their posterior. More specifically, given a training data set $\mathcal{D}$ and an unseen test sample $\boldsymbol{x}$ with class label $y$, we are interested in evaluating the predictive distribution $p(y|\boldsymbol{x}, \mathcal{D}) = \int p(y|\boldsymbol{x}, \boldsymbol{w}) p(\boldsymbol{w}|\mathcal{D}) \, \mathrm{d}\boldsymbol{w}$ . This integral requires to evaluate the posterior $p(\boldsymbol{w}|\mathcal{D})$, which involves the intractable marginal likelihood. A possible solution to this is to approximate the posterior with a more simple, tractable distribution $q(\boldsymbol{w})$ by optimization.

In this work, we incorporate the following approximately Bayesian methods which we use in our experiments to obtain weight uncertainty: Monte Carlo (MC) dropout (Gal & Ghahramani, 2016), Gaussian dropout (Wang & Manning, 2013; Kingma et al., 2015), Bayes by Backprop (Blundell et al., 2015), SWA-Gaussian (Maddox et al., 2019), and (although not Bayesian) deep ensembles (Lakshminarayanan et al., 2017). A short review of each of the methods can be found in Appendix A.2.

## 3 RELATED CALIBRATION METRICS

**Expected Calibration Error**  The expected calibration error (ECE) is one of the most popular calibration error metrics and estimates model calibration by binning the predicted confidences $\hat{p} = \max_c p(y = c \,|\, \boldsymbol{x})$ into $M$ bins from equidistant intervals and comparing them to average accuracies per bin (Naeini et al., 2015; Guo et al., 2017):

$$\text{ECE} = \sum_{m=1}^{M} \frac{|B_m|}{n} \Big| \text{acc}(B_m) - \text{conf}(B_m) \Big| \, , \tag{1}$$

with number of test samples $n$ and $\text{acc}(B)$ and $\text{conf}(B)$ denoting the accuracy and confidence of bin $B$, respectively. Several recent works have described severe pathologies of the ECE metric (Ashukha et al., 2020; Nixon et al., 2019; Kumar et al., 2019). Most notably, the ECE metric is minimized by a model constantly predicting the marginal distribution of the majority class which makes it impossible to directly optimize it (Kumar et al., 2018). Additionally, the ECE only considers the maximum class probability and ignores the remaining entries of the probability vector $\boldsymbol{p}(\boldsymbol{x})$.

**Adaptive Calibration Error**  Nixon et al. (2019) proposed the adaptive calibration error (ACE) to address the issue of fixed bin widths of ECE-like metrics. For models with high accuracy or overconfidence, most of the predictions fall into the rightmost bins, whereas only very few predictions fall into the rest of the bins. ACE spaces the bins such that an equal number of predictions contribute to each bin. The final ACE is computed by averaging over per-class ACE values to address the issue raised by Kull et al. (2019). However, this makes the metric more sensitive to the manually selected number of bins $M$ as the number of bins effectively becomes $C \cdot M$, with number of classes $C$. Using fixed bin widths, the numbers of samples in the sparsely populated bins is further reduced, which increases the variance of each measurement per bin. Using adaptive bins, this results in the lower confidence bins spanning a wide range of values, which increases the bias of the bin's measurement.

**Negative Log-Likelihood**  Deep models for classification are usually trained by minimizing the average negative log-likelihood (NLL):

$$\text{NLL} = \frac{1}{N} \sum_{i=1}^{N} - \log p(y = y_i \,|\, \boldsymbol{x}_i) \, . \tag{2}$$

The NLL is also commonly used as a metric for measuring the calibration of uncertainty. However, the NLL is minimized by increasing the confidence $\max_c p(y = c \,|\, \boldsymbol{x})$, which favors over-confident models and models with higher accuracy (Ashukha et al., 2020). This metric is therefore unable to compare the calibration of models with different accuracies and training a model by minimizing NLL does not necessarily lead to good calibration.

**Brier Score** The average Brier score is another popular metric for assessing the quality of predictive uncertainty and is defined as (Brier, 1950; Lakshminarayanan et al., 2017)

$$\text{BS} = \frac{1}{N} \sum_{i=1}^{N} \sum_{c=1}^{C} \left( \mathbf{1}(y_i = c) - p(y = c \,|\, \boldsymbol{x}_i) \right)^2 . \tag{3}$$

Similarly to the NLL, the Brier score favors high probabilities for correct predictions and low probabilities for incorrect predictions. Thus, models with higher accuracy tend to show a better Brier score, which makes the metric unsuitable for comparing the quality of uncertainty for models with different accuracies.

**Maximum Mean Calibration Error** Common recalibration methods are applied post-hoc, e.g. temperature scaling on a separate calibration set. Kumar et al. (2018) proposed the maximum mean calibration error (MMCE), a trainable calibration surrogate for the calibration error. It is defined as

$$\text{MMCE}^2(D) = \sum_{i,j \in D} \frac{\left( \mathbf{1}(\hat{y}_i = y_i) - \hat{p}_i \right) \left( \mathbf{1}(\hat{y}_j = y_j) - \hat{p}_j \right) k(\hat{p}_i, \hat{p}_j)}{m^2} \tag{4}$$

over batch $D \subset \mathcal{D}$ with batch size $m$, matrix-valued universal kernel $k$ and $\hat{y} = \arg \max_c p(y = c \,|\, \boldsymbol{x})$. Trainable calibration metrics are used in joint optimization with the negative log-likelihood

$$\arg \min_{\boldsymbol{w}} \sum_{D} \text{NLL}(D, \boldsymbol{w}) + \lambda \, \text{MMCE}(D, \boldsymbol{w}) . \tag{5}$$

Kumar et al. (2018) claim to have addressed the issue that the ECE is unsuitable for direct optimization due to its high discontinuity in $\boldsymbol{w}$. However, MMCE is also minimized by a model constantly predicting the marginal distribution of the classes. This leads to subpar logit temperature when training with MMCE and temperature scaling can further reduce miscalibration (Kumar et al., 2018).

## 4    UNCERTAINTY CALIBRATION ERROR

To give an insight into our general approach to measuring the calibration of uncertainty, we will first revisit the definition of perfect calibration of confidence (Guo et al., 2017) and show how this concept can be extended to calibration of our definition uncertainty.

Let $\hat{y} = \arg \max \boldsymbol{p}$ be the most likely class prediction of input $\boldsymbol{x}$ with confidence $\hat{p} = \max \boldsymbol{p}$ and true label $y$. Then, following Guo et al. (2017), *perfect calibration of confidence* is defined as

$$\mathbb{P}\left[ \hat{y} = y \,|\, \hat{p} = \alpha \right] = \alpha, \quad \forall \alpha \in [0, 1] . \tag{6}$$

That is, the probability of a correct prediction $\hat{y} = y$ given the prediction confidence $\hat{p}$ should exactly correspond to the prediction confidence. Instead of using only the probability of the predicted class, we use the entropy of $\boldsymbol{p}$ to express prediction uncertainty:

$$\mathcal{H}(\boldsymbol{p}) = - \sum_{c=1}^{C} p^{(c)} \log p^{(c)} . \tag{7}$$

Let
$$\boldsymbol{q}(k) := \left( \mathbb{P}[y = 1 | \arg \max \boldsymbol{p}(\boldsymbol{x}) = k], \dots, \mathbb{P}[y = C | \arg \max \boldsymbol{p}(\boldsymbol{x}) = k] \right) \tag{8}$$
be a probability vector of true marginal class probabilities for all inputs $\boldsymbol{x}$ predicted with class $k$. Consider the following example: Three i.i.d. inputs $\boldsymbol{x}_{1:3}$ in a binary classification task with ground truth labels $\{1, 1, 2\}$ have all been predicted with $\arg \max \boldsymbol{p}(\boldsymbol{x}_{1:3}) = 1$. Then, $\boldsymbol{q}(1) = \left( \frac{2}{3}, \frac{1}{3} \right)$. With this, we define a model to be perfectly calibrated if

$$\mathcal{H}(\boldsymbol{q}(k)) = \mathcal{H}(\boldsymbol{p} \,|\, \arg \max \boldsymbol{p} = k) \quad \forall k \in \{1, \dots, C\} . \tag{9}$$

From this, we derive an error metric for calibration of uncertainty:

$$\mathbb{E}_{\boldsymbol{p}} \left[ \left| \mathcal{H}(\boldsymbol{q}) - \mathcal{H}(\boldsymbol{p}) \right| \right] . \tag{10}$$

However, this metric and the use of the entropy as measure of uncertainty lacks interpretability, as the entropy scales with the number of classes $C$. This does not allow to compare the uncertainty or

the calibration of models trained on different data sets. Therefore, we propose to use the normalized entropy to scale the values to a range between 0 and 1:

$$\tilde{\mathcal{H}}(\boldsymbol{p}) := -\frac{1}{\log C} \sum_{c=1}^{C} p^{(c)} \log p^{(c)}, \quad \tilde{\mathcal{H}} \in [0, 1]. \tag{11}$$

We further increase interpretability and argue, that the normalized entropy should correlate with the model error. From Eq. (6) and Eq. (11), we define *perfect calibration of uncertainty* as

$$\mathbb{P}\big[\hat{y} \neq y \,|\, \tilde{\mathcal{H}}(\boldsymbol{p}) = \alpha\big] = \alpha, \quad \forall \alpha \in [0, 1]. \tag{12}$$

That is, in a batch of inputs that are all predicted with uncertainty of e. g. 0.2, a top-1 error of $20\,\%$ is expected. The confidence is interpreted as the probability of belonging to a particular class, which should naturally correlate with the model error of that class. This characteristic does not generally apply to entropy, and thus the question arises why entropy should correspond with the model error.

**Proposition 1.** *The normalized entropy (uncertainty) $\tilde{\mathcal{H}}(\boldsymbol{p})$ approaches the top-1 error in the limit of number of classes $C$ if the model $\boldsymbol{p}$ is well-calibrated.*

*Proof.*

$$\lim_{C \to \infty} \tilde{\mathcal{H}}(\boldsymbol{p}) = (1 - \hat{p}) \tag{13}$$

The top-1 error equals $(1 - \hat{p})$ if the model is perfectly calibrated in the sense of Eq. (6). For a detailed proof, see Appendix A.1. □

Thus, the normalized entropy gives us an intuitive and interpretable measure of uncertainty. If a model is perfectly calibrated, $\tilde{\mathcal{H}}$ corresponds to the top-1 error. We propose the following notion to quantify miscalibration of uncertainty:

$$\mathbb{E}_{\tilde{\mathcal{H}}}\Big[\big|\mathbb{P}\big[\hat{y} \neq y \,|\, \tilde{\mathcal{H}}(\boldsymbol{p}) = \alpha\big] - \alpha\big|\Big], \quad \forall \alpha \in [0, 1]. \tag{14}$$

We refer to this as Expected Uncertainty Calibration Error (UCE) and approximate with

$$\text{UCE} := \sum_{m=1}^{M} \frac{|B_m|}{n} \big|\text{err}(B_m) - \text{uncert}(B_m)\big|, \tag{15}$$

using the same binning scheme as in ECE estimation. The error per bin is defined as

$$\text{err}(B_m) := \frac{1}{|B_m|} \sum_{i \in B_m} \mathbf{1}(\hat{y}_i \neq y), \tag{16}$$

where $\mathbf{1}(\hat{y}_i \neq y) = 1$ and $\mathbf{1}(\hat{y}_i = y) = 0$. Uncertainty per bin is defined as

$$\text{uncert}(B_m) := \frac{1}{|B_m|} \sum_{i \in B_m} \tilde{\mathcal{H}}(\boldsymbol{p}_i). \tag{17}$$

**Properties of UCE**  The proposed UCE metric solves several problems of other metrics. First, the UCE is not zero for a model constantly predicting the marginal class distribution. Estimators of metrics with this pathology (e.g. ECE, MMCE) suffer from varying bias and therefore do not allow comparing miscalibration of different models (Ashukha et al., 2020; Vaicenavicius et al., 2019). In contrast to ACE, UCE is not highly sensitive to the numbers of bins and provides a consistent ranking of different models for the same classification task (see Fig. 1). Additionally, UCE can be used as a trainable regularizer in similar manner to MMCE. During training, we compute the UCE over mini-batches $D \subset \mathcal{D}$ and add it to the NLL training objective

$$\arg\min_{\boldsymbol{w}} \sum_{D} \text{NLL}(D, \boldsymbol{w}) + \lambda \, \text{UCE}(D, \boldsymbol{w}), \tag{18}$$

weighted by a factor $\lambda$. UCE is zero for an optimal model and thus does not penalize high confident predictions for models with high accuracy, which is a major disadvantage of plain entropy regularization (Pereyra et al., 2017). Predictions with low uncertainty, but high top-1 error are penalized whereas predictions with high accuracy are encouraged to have low uncertainty.

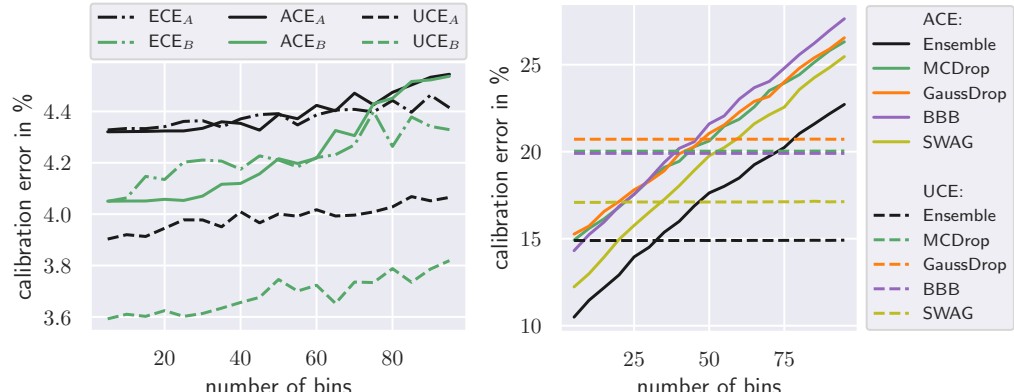

Figure 1: (Left) Calibration error values for two ResNet models $A$ and $B$ on CIFAR-10 test set. (Right) ACE and UCE values for different Bayesian methods on CIFAR-100 test set. ECE and ACE are very sensitive to the number of bins used in the estimator, not yielding a consistent ranking of the models. UCE is less sensitive to the bin number and ranks models consistently, allowing comparison of different models.

## 5 EXPERIMENTS

We evaluate the proposed uncertainty calibration error on mutli-class image classification on CIFAR-10 with ResNet-34 and on CIFAR-100 with ResNet-50 (He et al., 2016; Krizhevsky & Hinton, 2009). The feature extractor of ResNet is used as implemented in PyTorch 1.6 (Paszke et al., 2019) and the last linear layer is implemented using the different Bayesian approximations from § 2. All models were trained from random initialization. We employed early stopping at highest validation set accuracy. More details on the training procedure and a link to our source code can be found in Appendix A.3.

First, we compute the accuracies and all calibration error metrics from § 3 and the UCE on the test sets of CIFAR-10/100 for all models. We investigate the effect of the number of bins in the estimators of the metrics involving binning and analyze the ranking of different models under varying softmax temperature $\tau$, where $\boldsymbol{p} = \boldsymbol{\sigma}_{\mathrm{SM}}(\tau^{-1}\boldsymbol{f_w}(\boldsymbol{x}))$. Finally, we train a ResNet on CIFAR-10/100, SVHN, and Fashion-MNIST with added calibration error regularization as in Eq. (5) and (18). We compare UCE regularization ($\lambda = 10$) to regularization with MMCE ($\lambda = 10$) and confidence penalty $\arg\min \sum_D \mathrm{NLL}(D, \boldsymbol{w}) + \lambda \, \mathcal{H}(D, \boldsymbol{w})$ with $\lambda = 0.1$, which penalizes the entropy of the probability vector $\boldsymbol{p}$ of each prediction (Pereyra et al., 2017). We combine the regularization experiments with post-hoc calibration using temperature scaling (Guo et al., 2017).

Additionally, we analyze the utility of the normalized entropy as a measure of uncertainty and perform rejection and out-of-distribution (OoD) detection experiments using $\tilde{\mathcal{H}}$. We define an uncertainty threshold $\mathcal{H}_{\max}$ and reject all predictions from the test set where $\tilde{\mathcal{H}}(\boldsymbol{p}) > \mathcal{H}_{\max}$. A decrease in false predictions of the remaining test set is expected. To demonstrate the OoD detection ability, we provide images from CIFAR-100 to a deep model trained on CIFAR-10 (note that both CIFAR data sets have no mutual classes). In this experiment, we compose a batch of 100 random samples from the test set of the training domain and stepwise replace images with out-of-distribution data. In practice, it is expected that models are applied to a mix of known and unknown classes. After each step, we evaluate the mean batch uncertainty and expect, that the mean uncertainty monotonically increases as a function of the fraction of OoD data.

### 5.1 RESULTS

In this section, the results of the above mentioned experimental setups are presented and discussed.

**Comparison of Calibration Error Metrics** Table 2 shows test set accuracy and all calibration error results for all model/data set configurations. Without any post-hoc calibration, such as temperature scaling, all metrics provide the same ranking of the models. The deep ensemble and SWAG perform

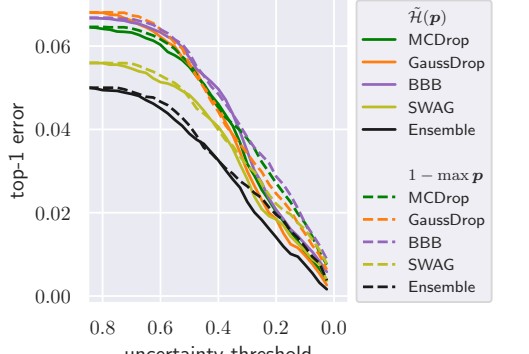 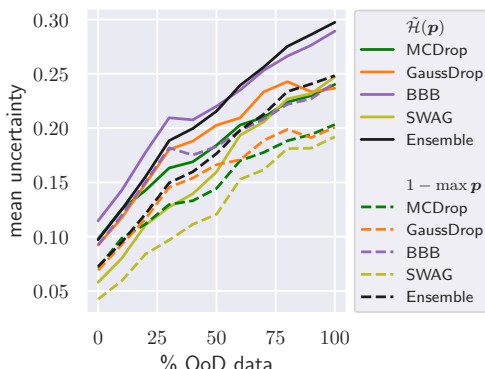

Figure 2: (Left) Rejection results on CIFAR-10 for decreasing uncertainty threshold comparing $\tilde{\mathcal{H}}(\boldsymbol{p})$ and $\max \boldsymbol{p}$ as uncertainty metric. In both cases, the top-1 error decreases strictly monotonically with decreasing threshold. (Right) Out-of-distribution detection for CIFAR-10 → CIFAR-100. The normalized entropy $\tilde{\mathcal{H}}(\boldsymbol{p})$ as measure of uncertainty can be used to robustly detect OoD data.

best in terms of test set accuracy and calibration of uncertainty. Brier score and NLL are both highly sensitive to the model accuracy, which is especially apparent on CIFAR-10. For the first three models with similar accuracy, the Brier scores differ only marginally. Thus, both the Brier score and the NLL are unsuitable for comparing the calibration of different models. Ashukha et al. (2020) propose to use the calibrated NLL at optimal temperature for model comparison. However, Fig. 5–6 plot the metrics over varying softmax temperature and show, that the models with highest accuracy have lowest Brier and NLL, regardless of the temperature. From this we deduce that both Brier and NLL should not be used for comparison of multi-class calibration, even at optimal temperature. The remaining metrics show consistent ranking before and after the point of optimal temperature. The metrics ECE, UCE and MMCE have a narrow region in which the optimal temperature for all models can be found. This allows comparison of calibration of models if they are all over- or underconfident. However, all metrics fail at comparing underconfident models to overconfident models (see model ranking left and right of optimal temperature in Fig. 6).

Fig. 1 shows the effect of the number of bins $M$ in the estimators of ECE, ACE and UCE. Both ECE and ACE are more sensitive to the number of bins and do not provide a consistent ranking of models under varying bin count. This is due to the fact that fewer bins are populated using $\tilde{\mathcal{H}}$ as uncertainty (cf. Fig. 10 in the appendix). This can be interpreted as possible downside of the UCE metric as the adaptive binning scheme of ACE explicitly addresses that. However, we argue that consistent ranking due to robustness against bin count results in a metric that is more useful in practice.

**Uncertainty Regularization** Tab. 1 shows results of ResNet-50 with SWAG trained on CIFAR-100. All regularization methods considerably reduce miscalibration compared to unregularized models. Plain entropy regularization is surprisingly effective on CIFAR-100; however, on CIFAR-10 (see Tab. 1), it increased miscalibration and is generally outperformed by MMCE and UCE regularization at optimal temperature. Therefore, when performing post-hoc temperature scaling, MMCE and UCE regularization is preferable to entropy regularization. UCE regularization can be interpreted as entropy penalization for predictions with low accuracy. As UCE is zero for an optimal model, it encourages a model to reach high accuracy.

**Rejection & OoD Detection** Fig. 2 (left) shows the top-1 error as a function of decreasing uncertainty threshold $\mathcal{H}_{\max}$ and (right) shows the mean batch uncertainty at increasing OoD data. Robust rejection of uncertain predictions and detection of OoD data based on the normalized entropy $\tilde{\mathcal{H}}(\boldsymbol{p})$ is possible and is generally more sensitive to OoD data than the confidence $\max \boldsymbol{p}$.

## 6 CONCLUSION

We have proposed to measure uncertainty based on the normalized entropy. From this, we derived the uncertainty calibration error; a new metric that avoids several pathologies of existing calibration errors. In our experimental evaluation, we focused on uncertainty from approximate Bayesian methods and deep ensembles. The UCE does not only consider the class with the highest probability and is not minimized by a constant model predicting the marginal class distribution. In contrast to the Brier score and NLL, it allows comparison of models with different accuracy. It is not sensitive to a varying number of bins and provides a consistent ranking of models. However, we follow the suggestion of Ashukha et al. (2020) and state that comparison of calibration for different models should only be done at optimal softmax temperature. Regularization with UCE during training reduces miscalibration and does not penalize high accuracy and predictions with justified high confidence. UCE regularization with temperature scaling often performed best in our experiments in terms of calibration. The normalized entropy itself is a useful measure of uncertainty and allows for robust rejection of uncertain predictions and detection of OoD data.

We hope to have provided a new useful metric for reliable evaluation of uncertainty estimation. UCE is easy to implement and interpretable as it expresses the discrepancy of the uncertainty from the model error, which increases the chance of being accepted by deep learning practitioners.

## 7 REBUTTAL

In this section, new or updated results during the rebuttal phase are presented. We identified the main question of the reviewers as to why/where our calibration metric (UCE) is beneficial compared to previous metrics (ECE, MMCE, ACE) and how normalized entropy is beneficial over vanilla entropy. We project the following main changes for the final manuscript:

- illustrate the advantage of normalized entropy over vanilla entropy
- add toy experiments to show that UCE is able to capture miscalibration where ECE, ACE and MMCE fail (see results below)
- add results from experiments on additional data sets (SVHN, Fashion-MNIST)

To make room for the following content, we propose the removal/reduction (approx. -1 page) of

- the review of Bayesian methods and refer to previous work instead,
- experiments regarding temperature (remove Table 1 and Figure 1)

Additionally, we will include all minor comments of the reviewers in the final manuscript.

### 7.1 UNCERTAINTY FROM NORMALIZED ENTROPY VS. ENTROPY

The advantages of normalized entropy over vanilla entropy in our definition of UCE are twofold: First, the domain of the metric does not scale with the number of classes $C$, which helps comparing the calibration of models trained on different data sets. Second, the value of the metric is more interpretable: Rejecting test samples where $\tilde{\mathcal{H}} < 0.2$ will result in a classification error $< 0.2$ for the remaining samples if the model is well-calibrated (see Fig. 2). We argue that using normalized entropy as uncertainty measure is as interpretable as $\max \boldsymbol{p}$, but avoids the pathologies of $\max \boldsymbol{p}$ when used in a calibration metric.

### 7.2 UCE VS. OTHER CALIBRATION METRICS

UCE is more reliable than ECE and MMCE because it is based on normalized entropy and incorporates the predictions of all classes. It is more robust than ACE, as it is significantly less sensitive to binning.

**Reliable Uncertainty Detection**    ECE and MMCE can be minimized by models with an uninformative, constant output (Ovadia et al., 2019). Given a data set of two classes, with 60 % class 1 and 40 % class 2, and a degenerated model that consistently predicts the marginal probabilities $\boldsymbol{p} = (0.6, 0.4)$. This leads to 60 % correctly classified samples for class 1 leading to perfect calibration scores for

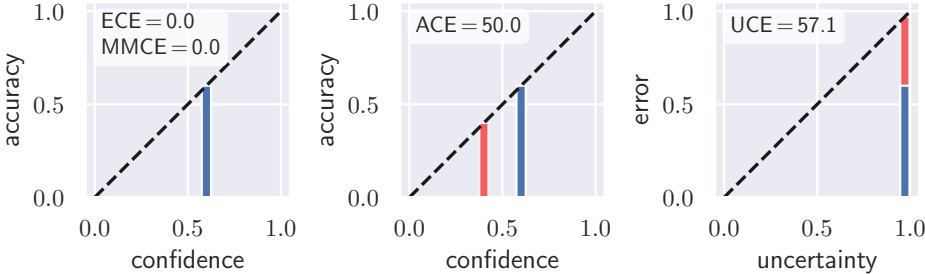

Figure 3: Calibration diagrams for a toy experiment with a degenerated model constantly predicting the marginal probabilities $\boldsymbol{p} = (0.6, 0.4)$ in a binary classification task. ECE and MMCE only consider $\max \boldsymbol{p}$ and fail at capturing the miscalibration of class 2 with $p(c = 2) = 0.4$, but $\mathrm{acc}(c = 2) = 0$. The red bars show the measured miscalibration. Uncertainty is given as normalized entropy. The left diagram is computed using ECE as MMCE does not involve binning.

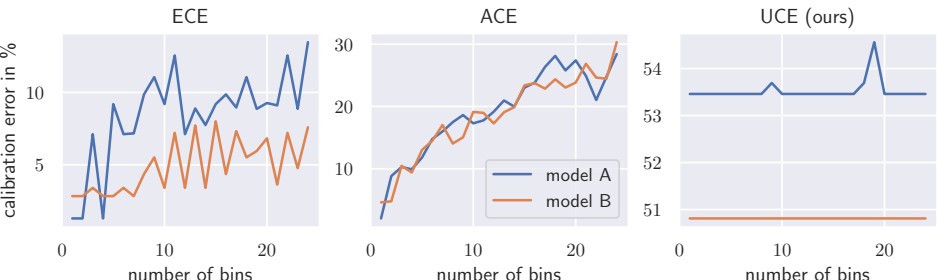

Figure 4: Toy experiment with two random models A and B in a binary classification task. UCE is less sensitive to number of bins used in the estimator and provides a consistent ranking of the models. For results from multi-class experiments, see Fig. 1.

ECE and MMCE. Class 2, however, is misclassified in 100% of the cases—and not 60 % as expected. The miscalibration of class 2 is not reflected in ECE or MMCE. ACE is computed class wise and is able to capture this miscalibration. UCE is based on the normalized entropy to determine uncertainty and therefore incorporates the predictions of all classes. Fig. 3 visualizes these results.

**Robustness to Varying Number of Bins**  From a reliable calibration metric, we would expect a constant ranking of two models in comparison, independent of varying calibration parameters. However, ACE is highly sensitive to the number of bins and produces arbitrary rankings of models. Fig. 4 highlights this by comparing two differently calibrated random models.

### 7.3   UCE REGULARIZATION

At optimal temperature (as suggested by Ashukha et al. (2020)), UCE and MMCE regularization considerably reduce miscalibration for all employed calibration metric outperforming entropy regularization, with UCE achieving highest accuracy on CIFAR-100, SVHN and Fashion-MNIST. (see Tab. 1). We want to stress out that UCE, in contrast to MMCE, was not specifically designed for the use as a calibration regularizer (Kumar et al., 2018).

**Why UCE Regularization Works**  UCE regularization works best when computed classwise (in similar manner to ACE): $\mathrm{UCE} = \frac{1}{C} \sum_{c=1}^{C} \mathrm{UCE}(c)$, where $\mathrm{UCE}(c)$ is computed for predictions of class $c$. Consider the following binary classification example: A batch with mainly samples from class 1 and few samples from class 2 are all predicted as class 1 with high confidence. NLL further pushes the confidence of the predictions to 1.0, favoring overconfidence, whereas UCE is only reduced if the confidence of the overconfidently false predictions is reduced.

Table 1: Revision of Table 1: Results from SWAG trained with entropy, MMCE and UCE regularization at optimal temperature (+T). We used the weighted MMCE (Kumar et al., 2018).

| Regularization | Dataset | Acc. | ECE | ACE | UCE | MMCE | Brier | NLL |
|---|---|---|---|---|---|---|---|---|
| unregularized | CIFAR-10 | **94.3** % | 3.8 % | 3.8 % | 3.6 % | 3.3 % | **0.10** | 0.28 |
| Entropy+T | CIFAR-10 | 94.1 % | 2.1 % | 4.2 % | 2.3 % | 1.1 % | **0.10** | 0.25 |
| MMCE+T | CIFAR-10 | 92.0 % | **0.4** % | **1.6** % | 0.8 % | **0.1** % | 0.12 | 0.24 |
| UCE+T (ours) | CIFAR-10 | 92.6 % | 0.5 % | **1.6** % | **0.7** % | 0.2 % | **0.10** | **0.21** |
| unregularized | CIFAR-100 | 68.3 % | 21.8 % | 22.0 % | 25.7 % | 18.3 % | 0.52 | 2.26 |
| Entropy+T | CIFAR-100 | 68.1 % | 2.9 % | 12.3 % | 3.7 % | 2.1 % | 0.44 | 1.41 |
| MMCE+T | CIFAR-100 | 67.7 % | **1.3** % | 11.0 % | 2.1 % | **0.5** % | 0.43 | 1.20 |
| UCE+T (ours) | CIFAR-100 | **70.9** % | 2.4 % | **10.4** % | **1.1** % | 1.2 % | **0.40** | **1.10** |
| unregularized | SVHN | 96.8 % | 2.10 % | 2.16 % | 1.89 % | 1.82 % | 0.05 | 0.19 |
| Entropy+T | SVHN | 96.9 % | 1.15 % | 2.31 % | 0.86 % | 0.74 % | 0.05 | 0.15 |
| MMCE+T | SVHN | **97.1** % | **0.27** % | **0.85** % | **0.35** % | 0.17 % | 0.05 | **0.12** |
| UCE+T (ours) | SVHN | **97.1** % | 0.38 % | 0.92 % | 0.38 % | **0.14** % | 0.05 | **0.12** |
| unregularized | F-MNIST | 94.7 % | 3.97 % | 3.96 % | 3.85 % | 3.60 % | 0.09 | 0.29 |
| Entropy+T | F-MNIST | 94.7 % | 1.86 % | 4.28 % | 2.13 % | 0.96 % | 0.09 | 0.24 |
| MMCE+T | F-MNIST | 94.7 % | 0.54 % | **1.40** % | 0.64 % | 0.17 % | **0.08** | **0.15** |
| UCE+T (ours) | F-MNIST | **94.8** % | **0.52** % | 1.75 % | **0.63** % | **0.11** % | **0.08** | 0.16 |

ACKNOWLEDGMENTS

Acknowledgments withheld.

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

## A    APPENDIX

### A.1    PROOFS

**Proposition 1.** *The normalized entropy (uncertainty) $\tilde{\mathcal{H}}(\boldsymbol{p})$ approaches the top-1 error in the limit of number of classes $C$ if the model $\boldsymbol{p}$ is well-calibrated.*

*Proof.* With Lemma 1 and $\hat{p} = \max \boldsymbol{p}$ we rewrite the normalized entropy as

$$\tilde{\mathcal{H}}(\boldsymbol{p}) = -\frac{\hat{p}\log\hat{p}}{\log C} - \frac{(1-\hat{p})\log\frac{1-\hat{p}}{C-1}}{\log C}\;. \tag{19}$$

Now, in the limit of number of classes $C$

$$\lim_{C\to\infty} \tilde{\mathcal{H}}(\boldsymbol{p}) = \lim_{C\to\infty} -\frac{(1-\hat{p})\log\frac{1-\hat{p}}{C-1}}{\log C} \tag{20}$$

$$= \lim_{C\to\infty} -(1-\hat{p})\left(\frac{\log(1-\hat{p})}{\log C} - \frac{\log(C-1)}{\log C}\right) \tag{21}$$

$$= (1-\hat{p}) \tag{22}$$

The top-1 error equals $(1-\hat{p})$ if the model is perfectly calibrated in the sense of Eq. (6).    □

**Lemma 1.** *Given a softmax output $\boldsymbol{p}$ with $C$ entries and the most likely prediction $\hat{y} = \arg\max\boldsymbol{p}$ with likelihood $\hat{p} = \max\boldsymbol{p}$. Then, the remaining entries $p_{i,i\neq\hat{y}}$ are approximately uniformly distributed with probability $\frac{1-\hat{p}}{C-1}$.*

*Proof.* This assumption is approximately correct (1) if $\hat{p} \to 1$ or (2) if $C \to \infty$. Let $\tilde{p}_j = p_i \;\forall i \neq \hat{y}$ and $\tilde{q}_j = \frac{(1-\hat{p})}{C-1}$. Note that $\tilde{p}$ and $\tilde{q}$ are not proper probability distributions as $\sum \tilde{p}_j = \sum \tilde{q}_j = (1-\hat{p})$.

(1) Consider $\text{KL}[\tilde{p}\|\tilde{q}]$ as $\hat{p}$ approaches 1:

$$\lim_{\hat{p}\to 1} \text{KL}\left[\tilde{p}\,\|\,\tilde{q}\right] = \lim_{\hat{p}\to 1} \sum_{j=1}^{C-1} \tilde{p}_j \log\frac{\tilde{p}_j}{\tilde{q}_j} \tag{23}$$

$$= \lim_{\hat{p}\to 1} \sum_{j=1}^{C-1} \tilde{p}_j \log\tilde{p}_j - \sum_{j=1}^{C-1} \tilde{p}_j \log\tilde{q}_j \tag{24}$$

$$= \lim_{\hat{p}\to 1} \sum_{j=1}^{C-1} \tilde{p}_j \log\tilde{p}_j - (1-\hat{p})\log\frac{(1-\hat{p})}{C-1} \tag{25}$$

$$= 0 \tag{26}$$

(2) Let $z_i$ be the logits of a model trained with L2 regularization. The magnitude of the logits $|z_i|$ cannot become arbitrary large and due to the normalizing nature of softmax

$$\lim_{C\to\infty} \frac{\exp z_i}{\sum_{j=1}^{C} \exp z_j} = \lim_{C\to\infty} \frac{1}{C}\;. \tag{27}$$

Alternatively, let $\boldsymbol{z} \in \mathbb{A}^C$ and $\boldsymbol{z}' \in \mathbb{B}^K$ be two logit vectors with $C < K$. If both models have been trained with L2 regularization, the magnitude of the logits $|z_i|, |z_i'|$ cannot become arbitrary large. More specifically, $\mathbb{A} = \mathbb{B} \subset \mathbb{R}$. Due to the normalizing nature of softmax, $\boldsymbol{z}'$ corresponds to a lower softmax temperature and as the temperature decreases with increasing number of classes, softmax approaches a uniform distribution (Jang et al., 2017).

□

## A.2 BAYESIAN DEEP LEARNING METHODS

In the following, we briefly describe common approximately Bayesian methods which we use in our experiments to obtain weight uncertainty.

**Monte Carlo Dropout**  One practical approximation of the posterior is variational inference with Monte Carlo (MC) dropout (Gal & Ghahramani, 2016). To determine model uncertainty, dropout variational inference is performed by training the model $\boldsymbol{f_w}$ with dropout (Srivastava et al., 2014) and using dropout at test time to sample from the approximate posterior distribution by performing $N$ stochastic forward passes per test sample (Gal & Ghahramani, 2016; Kendall & Gal, 2017). This is also referred to as MC dropout. In MC dropout, the final probability vector of the predictive distribution is computed by MC integration:

$$p(\boldsymbol{x}) = \frac{1}{N} \sum_{i=1}^{N} \boldsymbol{\sigma}_{\text{SM}} \left( \boldsymbol{f_{w_i}}(\boldsymbol{x}) \right). \tag{28}$$

**Gaussian Dropout**  Gaussian dropout was first proposed by Wang & Manning (2013) and linked to variational inference by Kingma et al. (2015). Dropout introduces Bernoulli noise during optimization and reduces overfitting of the training data. The resulting output $\boldsymbol{y}$ of a layer with dropout is a weighted sum of Bernoulli random variables. Then, the central limit theorem states, that $\boldsymbol{y}$ is approximately normally distributed. Instead of sampling from the weights and computing the resulting output, we can directly sample from the implicit Gaussian distribution of dropout

$$\boldsymbol{y} \sim \mathcal{N}(\mu_{\boldsymbol{y}}, \sigma_{\boldsymbol{y}}^2) \tag{29}$$

with

$$\mu_{\boldsymbol{y}} = \mathbb{E}[y_k] = \sum_j w_{j,k} x_j \,, \tag{30}$$

$$\sigma_{\boldsymbol{y}}^2 = \text{Var}[y_k] = p/(1-p) \sum_j w_{j,k}^2 x_j^2 \,, \tag{31}$$

using the reparameterization trick (Kingma et al., 2015)

$$y_j = \mu_j + \sigma_j \varepsilon_j \text{ with } \varepsilon_j \sim \mathcal{N}(0,1) \,. \tag{32}$$

Gaussian dropout is a continuous approximation to Bernoulli dropout, and in comparison it will better approximate the true posterior distribution and is expected to provide improved uncertainty estimates (Louizos & Welling, 2017). To obtain the final probability vector $p(\boldsymbol{x})$, we again use MC integration with $N$ stochastic forward passes.

The dropout rate $p$ is now a learnable parameter and does not need to be chosen carefully by hand. In fact, $p$ could be optimized w.r.t. uncertainty calibration, scaling the variance of the implicit Gaussian of dropout. A similar approach was presented by Gal et al. (2017) using the Concrete distribution (Maddison et al., 2016; Jang et al., 2017). However, we focus on metrics for measuring calibration and therefore fix $p$ in our subsequent experiments.

**Bayes by Backprop**  Blundell et al. (2015) assume a Gaussian distribution with diagonal covariance matrix as variational posterior $q(\boldsymbol{w}|\boldsymbol{\theta})$, parameterized by mean $\boldsymbol{\mu}$ and standard deviation $\boldsymbol{\sigma}$, where $\boldsymbol{\theta} = \{\boldsymbol{\mu}, \boldsymbol{\sigma}\}$. A sample of the weights can be obtained by sampling a multivariate unit Gaussian and shift it by $\boldsymbol{\mu}$ and scale it by $\boldsymbol{\sigma}$. Then, the network is trained by minimizing

$$\mathcal{L}(\boldsymbol{\theta}) = \text{KL}[q(\boldsymbol{w}|\boldsymbol{\theta})\|p(\boldsymbol{w})] - \mathbb{E}_q[\log p(\mathcal{D}|\boldsymbol{w})] \,. \tag{33}$$

In case of a zero mean Gaussian prior, the first term can be implemented by weight decay. In contrast to Gaussian dropout, which operates on the implicit distribution of the activations, Bayes by Backprop (BBB) directly operates on the weights. This doubles the number of trainable parameters in practice. MC integration is used to obtain the final probability vector $p(\boldsymbol{x})$.

**SWA-Gaussian**  Stochastic weight averaging (SWA) uses stochastic gradient descent steps around a local loss optimum of a trained network and averages the weights $w_{\text{SWA}} = \frac{1}{T} \sum_{i=1}^{T} w_i$ of the model from each step $i$ (Izmailov et al., 2018). This explores the loss landscape and averaging helps to find a better weight estimate than converging to a single local optimum. SWA-Gaussian (SWAG) is closely related to Bayes by Backprop (Maddox et al., 2019). It assumes a Gaussian distribution with diagonal covariance matrix as approximate variational posterior. Instead of using backpropagation to directly optimize $\mu$ and $\sigma$, it fits a Gaussian by using $\mu = w_{\text{SWA}}$ and

$$\Sigma_{\text{diag}} = \text{diag}(\overline{w^2} - w_{\text{SWA}}^2), \qquad \overline{w^2} = \frac{1}{T} \sum_{i=1}^{T} w_i^2 . \tag{34}$$

This doubles the number of parameters at test time. The approximate Gaussian posterior results to $\mathcal{N}(w_{\text{SWA}}, \Sigma_{\text{diag}})$ and MC integration with samples $w_i \sim \mathcal{N}(w_{\text{SWA}}, \Sigma_{\text{diag}})$ is used to compute the final probability vector $p(x)$.

**Deep Ensembles**  Training multiple randomly initialized copies of a deep network by performing maximum posterior estimation and ensembling them to get multiple predictions for a single input is not a variational inference method. However, they have been reported to produce surprisingly useful uncertainty estimates in practice that are better calibrated (Lakshminarayanan et al., 2017). Deep ensembles considerably increase the number of parameters at train and test time. We use deep ensembles as non Bayesian baseline for uncertainty estimation.

### A.3  TRAINING SETTING

The base model implementations from PyTorch 1.6 (Paszke et al., 2019) are used and trained with following settings:

- Adam optimizer with initial learn rate of 3e-4 and $\beta_1 = 0.9, \beta_2 = 0.999$ and mini-batch size of 256 (Kingma & Ba, 2015)
- weight decay of 1e-6
- negative-log likelihood (cross entropy) loss
- reduce-on-plateau learn rate scheduler (patience of 20 epochs) with factor of $0.1$
- additional validation set is randomly extracted from the training set ($5,000$ samples)
- ResNet-34 for CIFAR-10 and ResNet-50 for CIFAR-100 experiments
- only the last linear layer is implemented in a Bayesian manner for MC dropout, Gaussian dropout, BayesByBackprop and SWAG
- the deep ensemble comprises 3 fully individually trained networks
- $N = 25$ forward passes were used Monte Carlo integration
- in MC dropout and Gaussian dropout, a dropout rate of $p = 0.2$ was used
- in SWAG, a learn rate of 3e-6 was used during weight averaging

Our code is available at: https://github.com/link-withheld.

## A.4 ADDITIONAL RESULTS

| Bayes | Dataset | Accuracy | ECE | ACE | UCE | MMCE | Brier | NLL |
|---|---|---|---|---|---|---|---|---|
| MC Drop | CIFAR-10 | 93.6 % | 4.3 % | 4.3 % | 4.0 % | 3.8 % | 0.11 | 0.31 |
| Gauss Drop | CIFAR-10 | 93.2 % | 4.4 % | 4.4 % | 4.1 % | 3.8 % | 0.11 | 0.31 |
| BBB | CIFAR-10 | 93.3 % | 4.6 % | 4.6 % | 4.4 % | 4.1 % | 0.11 | 0.34 |
| SWAG | CIFAR-10 | 94.4 % | 3.7 % | 3.7 % | 3.5 % | 3.5 % | 0.09 | 0.28 |
| Ensemble | CIFAR-10 | 95.0 % | 3.2 % | 3.2 % | 3.0 % | 2.8 % | 0.08 | 0.22 |
| MC Drop | CIFAR-100 | 66.9 % | 24.4 % | 24.5 % | 27.9 % | 20.6 % | 0.55 | 2.55 |
| Gauss Drop | CIFAR-100 | 66.5 % | 24.5 % | 24.7 % | 28.2 % | 20.7 % | 0.56 | 2.64 |
| BBB | CIFAR-100 | 65.1 % | 24.9 % | 25.1 % | 28.9 % | 20.9 % | 0.57 | 2.51 |
| SWAG | CIFAR-100 | 68.3 % | 21.8 % | 22.1 % | 25.7 % | 18.3 % | 0.52 | 2.26 |
| Ensemble | CIFAR-100 | 72.5 % | 19.2 % | 19.4 % | 22.5 % | 16.1 % | 0.45 | 1.82 |

Table 2: Classification accuracy and calibration error results for different models on CIFAR-10/100. We used $M = 15$ bins where necessary. Here, all metrics provide the same ranking of models.

## A.5 ADDITIONAL FIGURES

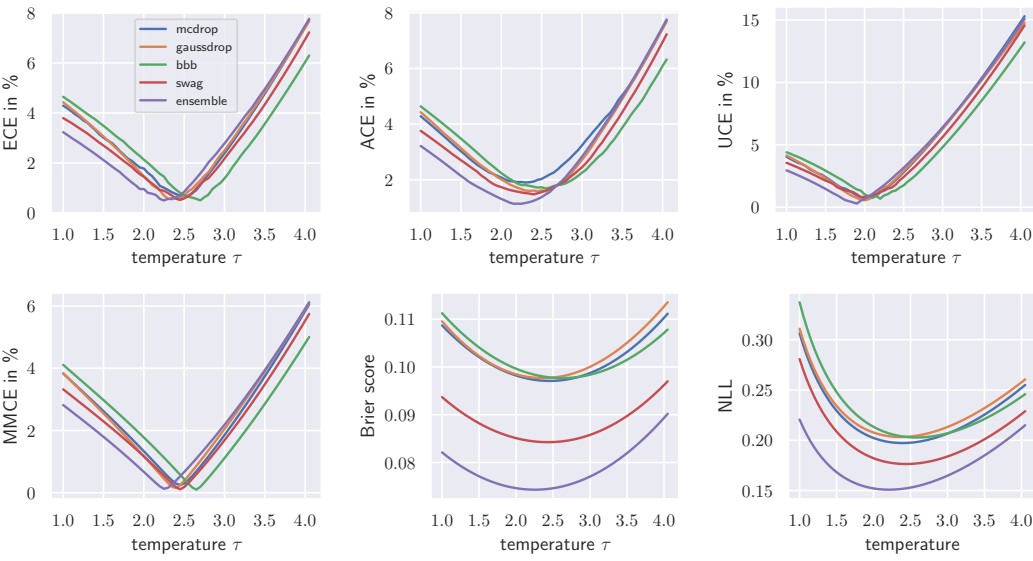

Figure 5: Calibration error vs. softmax temperature on CIFAR-10. All metrics provide inconsistent ranking of models over $\tau$. The metrics ECE, UCE and MMCE have a narrow region in which the optimal temperature for all models can be found. They show more a consistent ranking before and after the point of optimal temperature. This allows comparison of calibration of models if they are all over- or under-confident. However, all metrics fail at comparing underconfident models to overconfident models. Even at optimal temperature, Brier score and NLL fail at comparing calibration of models with different accuracy, as the metrics are always lower for models with better accuracy.

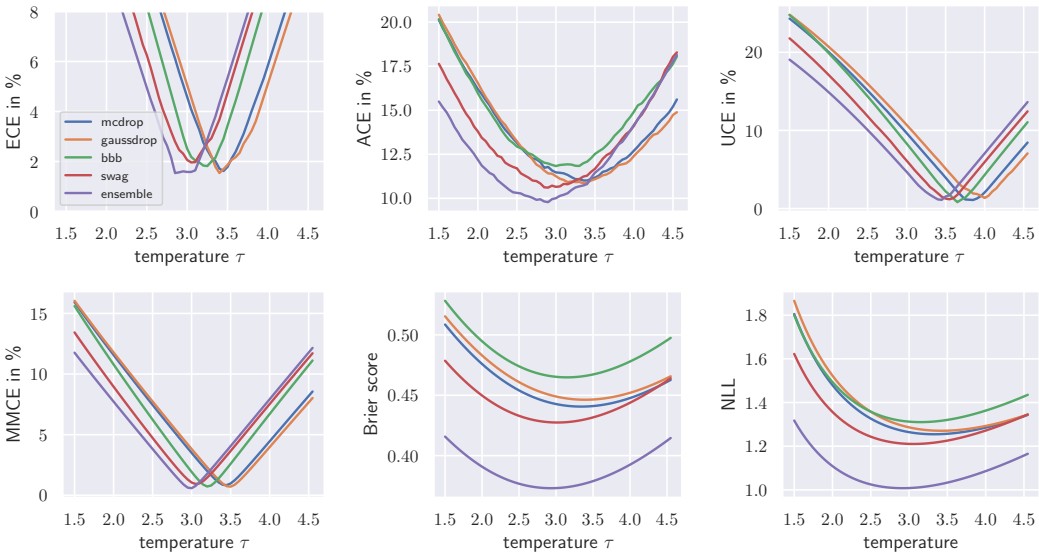

Figure 6: Calibration error vs. softmax temperature on CIFAR-100.

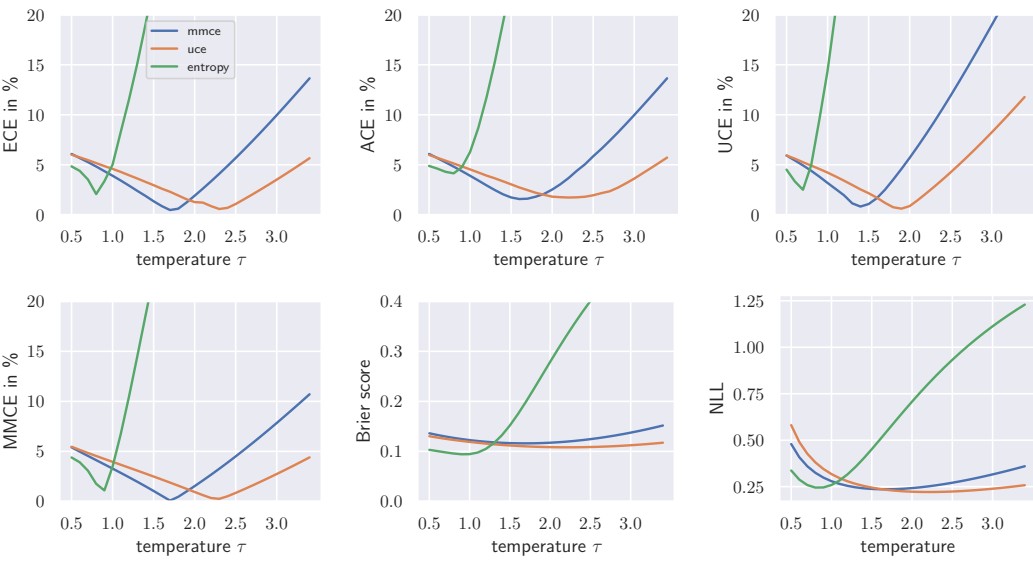

Figure 7: Calibration error vs. softmax temperature from SWAG trained with different regularization on CIFAR-10. Both MMCE and UCE regularization lead to less overconfident models and reduce miscalibration (optimal temperature is closer to $\tau = 1$). Entropy regularization leads to underconfident models and is not as effective as MMCE and UCE regularization on CIFAR-10. MMCE and UCE regularization at optimal temperature outperform entropy regularization at optimal temperature for all metrics except Brier score.

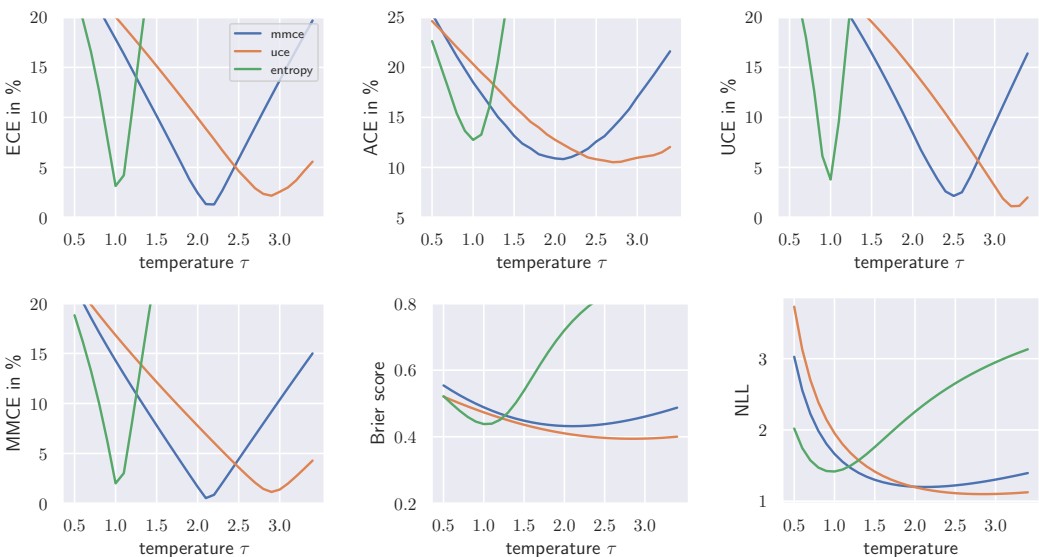

Figure 8: Calibration error vs. softmax temperature from SWAG trained with different regularization on CIFAR-100. In this experiment, entropy regularization without temperature scaling ($\tau = 1$) was surprisingly effective and outperforms MMCE and UCE regularization. However, at optimal temperature both MMCE and UCE regularization outperform entropy regularization for all metrics.

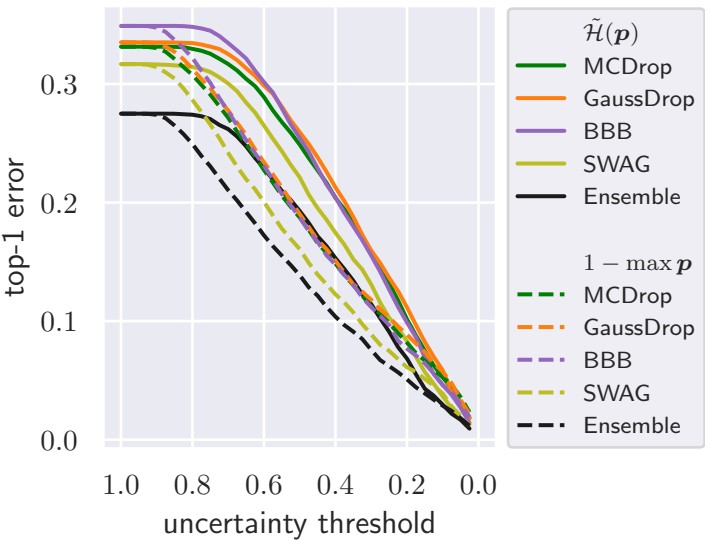

Figure 9: Rejection results on CIFAR-100 for decreasing uncertainty threshold comparing $\tilde{\mathcal{H}}(\boldsymbol{p})$ and $\max \boldsymbol{p}$ as uncertainty metric. In both cases, the top-1 error decreases monotonically with decreasing threshold.

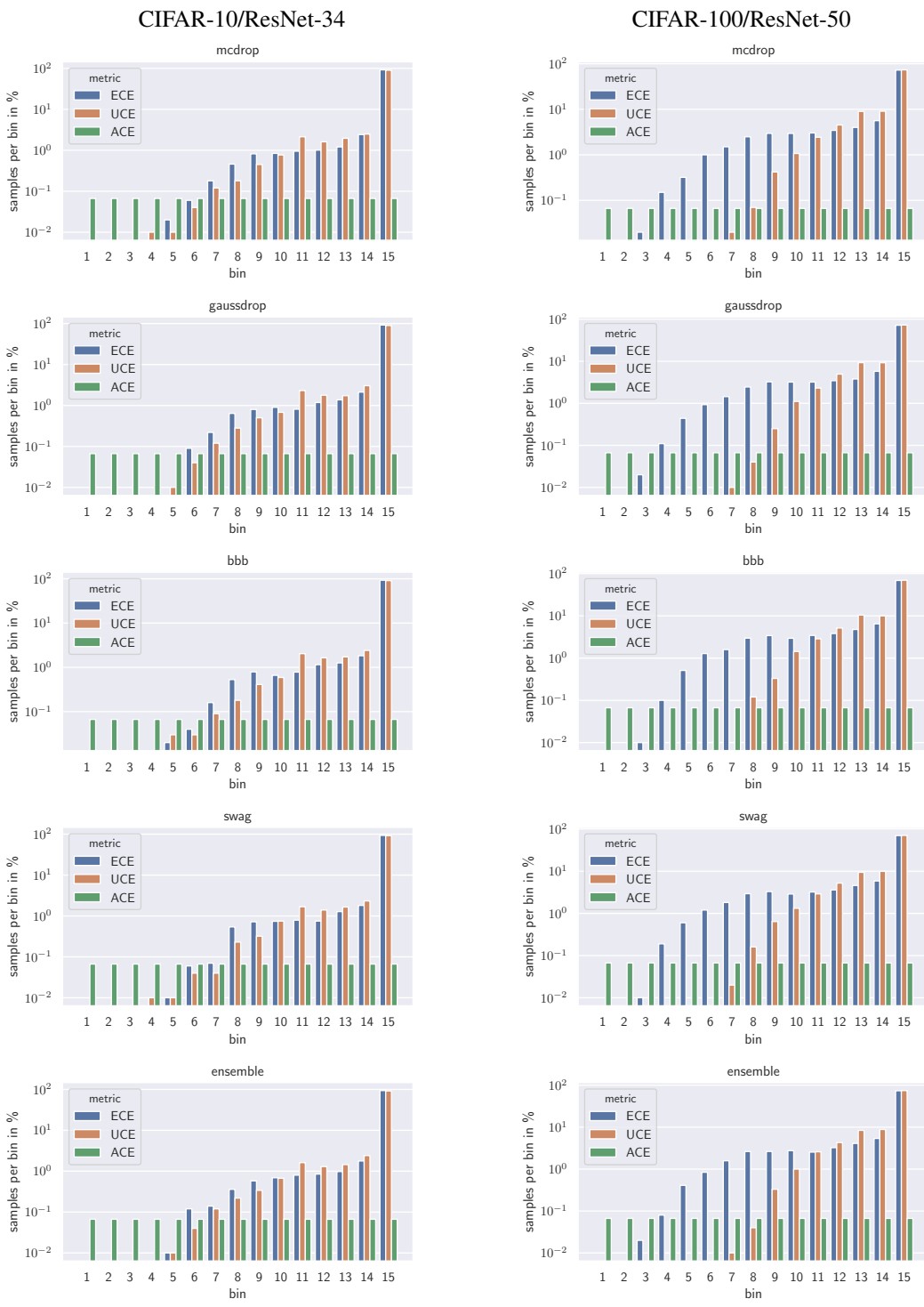

Figure 10: Binning estimator sample distribution for ResNet-34 on CIFAR-10 (left) and for ResNet-50 on CIFAR-100 (right) with $M = 15$ bins. ECE and UCE use fixed bin widths and ACE uses an adaptive binning scheme. The number of samples per bin for ECE and UCE are similar on CIFAR-10. On CIFAR-100, UCE favors fewer bins, which makes UCE more insensitive to the total number of bins. Due to the adaptive binning, ACE is highly sensitive to the bin count.

