# OpenReview forum: "Uncertainty Calibration Error: A New Metric for Multi-Class Classification"
_ICLR.cc/2021/Conference — Reject_

### Official Review · AnonReviewer3 · 2020-10-28
**Reviewer3**

**Rating:** 5
**Confidence:** 3

**Review:**

The work addresses an important problem in the study of uncertainty estimation: how does one compare model uncertainty at differing accuracy levels? The work proposes a novel uncertainty metric, relates this to existing methods and provides robust evaluation of the various merits of this approach. The paper is easy to follow.

I have the following concerns with the work:
1. Regarding the use of UCE as a regularizer: how is the described behaviour of UCE different from the NLL loss? The NLL loss should penalize highly confident incorrect predictions and strives for confident predictions in high accuracy. What does the UCE regularizer add here?  Can table 2 include a non-regularized baseline as well to study this? In appendix A.3. it is said that UCE performs on par without regularization; then what is the point of proposing UCE as a regularizer?
2. What is the point of proposing the use of the normalized entropy as a thresholding factor for OOD detection? It seems that vanilla entropy would behave exactly the same. Is this considered to be a novel contribution in this work?
3. Why is figure 2 (right) completely flat for UCE? Are there values not shown here where calibration error does change? Perhaps this should be included in the plot.
4. Am I correct to say that max-p-based metrics might be preferable in very large-class problems such as language models? The paper does not discuss the computational tradeoffs of the method, and I believe this should be included.
5. It appears that the experiment section does not provide much evidence  that this metric is favourable in selecting the best model for a downstream task where uncertainty is needed. This could be evaluated by e.g. an active learning problem. I believe it makes sense to include such an experiment. Right now, Table 1 and the accompanying discussion does not convince me that UCE is somehow more beneficial.

Overall, the work has merit and of interest to the community. However, the proposal of the use of the metric as a regulariser and a OOD scoring function seems unproductive and if so, distracts from the core contribution. This core contribution is understudied in the work. The work would benefit from more analysis into the computational tradeoffs, and evaluation of the signal that the proposed metric provides on model selection for downstream uncertainty tasks.

Nitpick:
- Table 1 could benefit from a vertical bar between the two datasets to clarify that the numbers are not comparable.

Update:
Although the paper has improved, I still vote for rejection. The new insight of binary-classwise v/s multiclass UCE as a regularizer seems poorly explored in the paper and would benefit from closer study. This appears to be the basis of the improved results in table 1.

---

> ### Author Response · Authors · 2020-11-15
> **Benefits of UCE**
>
> Dear AnonReviewer3, thank you for recognizing the importance of our work and for your constructive feedback. In the following we will address your concerns point by point.
>
> Ad 1.: First, we want to stress out that the use of UCE as a regularizer is not our main finding, but rather an interesting fact. UCE regularization works best when computed classwise (in similar manner to ACE). Consider the following example: A batch with mainly samples from class 1 and few samples from class 2 are all predicted as class 1 with high confidence. Increasing the confidence of all predictions further reduces the NLL, whereas UCE is only reduced if the confidence of the overconfidently false predictions is reduced. Moreover, when compared at optimal temperature (as suggested by Ashukha et al. (2020)), UCE regularization is at least as good as MMCE reg. and outperforms entropy reg. (see Fig. 6 in appendix). We will add a table with metric values at optimal temperature to the main text to highlight the benefits of UCE regularization.
>
> Ad 2.: Thank you for your question. It is correct that the rejection experiments could have been conducted with vanilla entropy. Our experiments aim at highlighting the interpretability of UCE/normalized entropy: Rejecting test samples where $ \tilde{\mathcal{H}} > 0.2 $ will result in a classification error $< 0.2$ for the remaining samples if the model is well-calibrated (see Fig. 9 & 2). We argue that using normalized entropy as uncertainty measure is as interpretable as max p, but avoids the pathologies of max p when used in a calibration metric. We will make this clearer in the text.
>
> Ad 3.: Thank you for pointing that out. The figure was created by incrementing the number of bins in steps of 5 from 5 to 100. We will recreate this figure using an increment of 1 and a smaller y axis range. Small fluctuations of the UCE values should then become visible. However, this does not change our finding that UCE is not sensitive to the number of bins and provides a consistent ranking of the models.
>
> Ad 4.: This is an interesting comment, indeed. For very large-class problems,  max p based metrics and our metric should be equivalent, but max p based metrics are computationally more efficient. We will follow your suggestion and discuss this in our upcoming revision.
>
> Ad 5.: Thank you for your assessment of our experiments. We already conducted additional toy experiments that better highlight the benefits of UCE. The experiments show that UCE can measure miscalibration where the other metrics fail. We are currently working hard to also include an active learning experiment. We hope that this will convince you of our work.
>
> We hopefully have considered all of your concerns and welcome any further discussion.
>
> References
>
> *see paper*

---

> > ### Comment · AnonReviewer3 · 2020-11-20
> > **regarding NLL**
> >
> > Re 1.
> > It seems that this is an incorrect characterisation of NLL. NLL heavily penalizes overconfident incorrect predictions, see the following:
> > ```
> > import jax
> > from jax import nn
> > from jax import numpy as np
> >
> > pred_a = np.asarray([[0.8,0.2]]*10)
> > pred_b = np.asarray([[0.9999,1-0.9999]]*10)
> > true = np.asarray([1] * 9 + [0])
> >
> > def cross_entropy(logprobs, target_class):
> >   nll = np.take_along_axis(logprobs, np.expand_dims(target_class, axis=1), axis=1)
> >   ce = -np.mean(nll)
> >   return ce
> >
> > 'less confident', float(cross_entropy(np.log(pred_a), true)), 'confident', float(cross_entropy(np.log(pred_b), true))
> > ```
> > ('less confident', 1.470808506011963, 'confident', 8.28931713104248)
> >
> > I'm not sure what you mean with classwise. Do you mean binary multi-class prediction rather than categorical prediction?

---

> > > ### Author Response · Authors · 2020-11-20
> > > **Response to NLL Concerns**
> > >
> > > Dear AnonReviewer3, thank you for this helpful code example.
> > >
> > > 1. We think that you may have swapped the labels in line 7. Let us prove a specific code example:
> > > ```
> > > pred_a = np.asarray([[0.9,1-0.9]]*9 + [[0.8,1-0.8]])
> > > pred_b = np.asarray([[0.99,1-0.99]]*9 + [[0.89,1-0.89]])
> > > true = np.asarray([0] * 9 + [1])
> > > 'less confident', float(cross_entropy(np.log(pred_a), true)), 'confident', float(cross_entropy(np.log(pred_b), true))
> > > ```
> > > ('less confident', 0.2557682553354537, 'confident', 0.22977279358712344)
> > > I.e. NLL further pushes the confidence of the predictions to 1.0, favoring overconfidence.
> > > 2. By classwise we mean $ \frac{1}{C} \sum_{c=1}^{C} \mathrm{UCE}_{c} $, where $ \mathrm{UCE} _{c} $ is computed for samples of class $ c $. We will add sentences on this to our revision.

---

> > > > ### Comment · AnonReviewer3 · 2020-11-20
> > > > **Good catch!**
> > > >
> > > > Thank you for spotting the mistake! It seems that when I flip the labels in my example, the (over)confident predictions still have the higher loss:
> > > > ```true = np.asarray([0] * 9 + [1]```
> > > > ('less confident', 0.36177298426628113, 'confident', 0.9211241006851196)
> > > >
> > > > (Although I would agree that your example is more realistic).
> > > >
> > > > I wonder how UCE compares here, but would you agree that this example indicates that the difference between NLL and UCE as a regulariser is perhaps more subtle than described in the paper?

---

> > > > > ### Author Response · Authors · 2020-11-20
> > > > > **UCE**
> > > > >
> > > > > In this specific binary classification example with $ C=2 $, our Proposition 1 does not hold and we would rather suggest to use max p instead of $ \tilde{\mathcal{H}} $ (which is effectively regularization with classwise ECE (Kull et al., 2019)). Then, we would have
> > > > > ```
> > > > > print(cross_entropy(pred_a, true))
> > > > > print(cross_entropy(pred_b, true))
> > > > >
> > > > > print(classwise_ece(pred_a, true))
> > > > > print(classwise_ece(pred_b, true))
> > > > > ```
> > > > > ```
> > > > > 0.2558
> > > > > 0.2298
> > > > >
> > > > > 0.0100
> > > > > 0.0800
> > > > > ```
> > > > > However, in a multiclass scenario, the same holds for UCE (plus the benefits of UCE ).

---

> > ### Comment · AnonReviewer3 · 2020-11-20
> > **re 2**
> >
> > Re 2: This is a good point, agreed. The accompanying proof of this fact is also of value here. I do think it's helpful to acknowledge this in the paper.
> >
> > Re 3: That sounds good! Perhaps exploring the limit all the way to 1 bins can help clarify that there is a minimum number of bins required to correctly estimate the metric.
> >
> > Re 4: Great, I think that insight is helpful for the reader.
> >
> > Re 5: This seems to me to be the missing piece of the story. If you propose that UCE should be the go-to metric for comparing different methods/models on calibration error,  it seems important to demonstrate practical settings in which UCE is more informative than alternative metrics for model selection.

---

> > > ### Author Response · Authors · 2020-11-20
> > > **Response**
> > >
> > > Thank you again for your valuable feedback. We are currently working hard on a revision of our manuscript and would appreciate an updated rating if we were able to address all your concerns.

---

> > > ### Author Response · Authors · 2020-11-24
> > > **Additional Experiments**
> > >
> > > Dear AnonReviewer3, please see the rebuttal version for the added toy experiments. We are still working your suggested active learning experiment. However, due to the time constraints of the rebuttal phase, we were not able to provide these results yet. We will include the results in a possible camera-ready version.

---

### Official Review · AnonReviewer1 · 2020-10-29
**Good proposal but an enhanced set of experiments are required**

**Rating:** 4
**Confidence:** 3

**Review:**

This paper proposes a new calibration error measurement named UCE (Uncertainty Calibration Error) for deep classification models. It consists in doing a calibration in order to achieve "perfect calibration" (i.e., the uncertainty provided is equivalent to the classification error at all levels in [0, 1]), relying on normalized entropy for multiclass classification. This UCE is well justified for classification problems with several classes to process, where the entropy is demonstrated to be asymptotically equivalent to the classification (top-1) error. A point with this UCE metric is that is has some interpretability properties in terms of its value, and is said to be robust to the number of bins used.

The proposed metric is well explained, and justified, although I am wondering how well stands the assumption that the normalized entropy approaches the top-1 error for reasonable number of classes (e.g. C=10, as with CIFAR-10, or C=100, as with CIFAR-100). The properties presented are interesting.

If found the experiments to be well aligned to evaluate the approach, although limited in terms of dataset used (only CIFAR-10 and CIFAR-100), a greater variety of datasets would be more convincing in the of overall good performances of the approach, especially if datasets with a varied number of classes can be tested. Moreover, looking at the results in detail (Table 1), UCE does not appear to be particularly strong, having a worse calibration than ECE and ACE on CIFAR-10, but slightly better on CIFAR-100, assuming that we want it to be increased to reach the real error rate obtained. Moreover, the presentation of the results in Table 1 is messy: it gets difficult to match the calibration error with the accuracy, providing the classification error instead of accuracy would help to make a direct comparison with calibration error. Moreover, why the last two columns in Table 1 (Brier and NLL) are provided as floating-point values instead of percentages as with the other columns. That's unnecessary confusion that should be fixed.

Overall, I found the paper to be correct, relatively well written. I think that more room should have been given to experimentations, like with other datasets and with more space for OoD rejection and detection. Conversely, I am not sure of the relevance of providing all the detailed information on Bayesian methods in the second part of Sec. 2. It can be presented in a more concise way, as it uses a lot of space to explain well-known approaches.

In terms of potential impact of that paper, I still need to be convinced. What can tell me that this is just not yet another calibration metric. I think that the paper can have been made stronger on that aspect.

---

> ### Author Response · Authors · 2020-11-13
> **Reducing Confusion**
>
> Dear AnonReviewer1, we thank you very much for your thorough review of our work and the positive comments. In the following we try to address all your concerns point by point.
>
> > If found the experiments to be well aligned to evaluate the approach, although limited in terms of dataset used (only CIFAR-10 and CIFAR-100), a greater variety of datasets would be more convincing in the of overall good performances of the approach, especially if datasets with a varied number of classes can be tested.
>
> Thank you for mentioning our limited use of data sets. As of writing this, we conduct additional experiments on SVHN and Fashion-MNIST and provide the results in our revised manuscript. We expect these results to be aligned with the results on CIFAR-10/100.
>
> > Moreover, looking at the results in detail (Table 1), UCE does not appear to be particularly strong, having a worse calibration than ECE and ACE on CIFAR-10, but slightly better on CIFAR-100, assuming that we want it to be increased to reach the real error rate obtained.
>
> After reading your comment, we realized that we do not present our results clearly and comprehensibly. Table 1 shows the considered metrics on uncalibrated models and we do not expect any metric to reflect the model error at this point. Moreover, we argue that for well-calibrated models, the normalized entropy (as notion of uncertainty) should reflect the model error. We do not argue that the value of the metric itself reflects the error; it rather shows the deviation of our assumption of perfect calibration, see Eq. (22). We will add details to the caption of Table 1 and rewrite the sentences discussing the results in the main text. Thank you very much for drawing our attention to this.
>
> > Moreover, the presentation of the results in Table 1 is messy: it gets difficult to match the calibration error with the accuracy, providing the classification error instead of accuracy would help to make a direct comparison with calibration error.
>
> We think that this issue is mainly addressed above. We provide the accuracy as we do not expect the calibration metrics to equal the error in Table 1; and providing the classification error could further add confusion. However, we will rework Table 1 as suggested by AnonReviewer3 and hope that this, in addition to a more detailed description of the results, will meet your expectations.
>
> > Moreover, why the last two columns in Table 1 (Brier and NLL) are provided as floating-point values instead of percentages as with the other columns. That's unnecessary confusion that should be fixed.
>
> Thank you for pointing that out. We mainly followed related work where ECE-like metrics were provided as percentages, and Brier and NLL were provided as floating-point values, e.g. see (Kumar et al., 2018). Moreover, as pointed out by AnonReviewer4, NLL and Brier are strictly proper scoring rules and have to be decomposed in order to be directly comparable to other calibration metrics. We will visually seperate the NLL and Brier scores from the other calibration metrics using a vertical bar to reduce this confusion (see also answer to AnonReviewer4).
>
> > Conversely, I am not sure of the relevance of providing all the detailed information on Bayesian methods in the second part of Sec. 2. It can be presented in a more concise way, as it uses a lot of space to explain well-known approaches.
>
> Thank you for this suggestion. We will shorten the description of the Bayesian methods and use the free space for the results of the new experiments.
>
> > In terms of potential impact of that paper, I still need to be convinced. What can tell me that this is just not yet another calibration metric. I think that the paper can have been made stronger on that aspect.
>
> Many recent papers have highlighted the need for an appropriate calibration metric (see our response to AnonReviewer2). Our metric reliably detects miscalibration as it avoids various pathologies of other metrics. We are convinced that our work is a valuable contribution to the community. To respond to your comment, we will rephrase the conclusion of our paper to make it stronger. In addition to the expected results from the new experiments, we hope to convince you and would be very grateful if you would update your rating accordingly.
>
> References
>
> *see paper*

---

### Official Review · AnonReviewer2 · 2020-10-29
**Well-motivated metric for uncertainty calibration; novelty is unclear**

**Rating:** 6
**Confidence:** 3

**Review:**

Update: After reading the other reviews and responses, and in light of the authors' updates to the paper, I have increased my score to a 6.

This paper proposes a new metric for uncertainty calibration, based on comparing the entropy of the marginal class probabilities conditioned on predicted class with the entropy of the predicted probabilities. The metric avoids the failure mode of ECE, where predicting the relative frequencies of classes results in perfect calibration, and can be used as a regularizer in a loss function. The paper demonstrates that regularization with UCE yields better-calibrated uncertainty on CIFAR predictions without sacrificing accuracy.

The paper is well-written and well-motivated. I’m uncertain as to its novelty. In particular, entropy as a basis for uncertainty estimation is well-explored (and was used as a baseline in (Lakshminarayanan et al., 2017) as well as Jie Ren et al., “Likelihood ratios for out-of-distribution detection,” NEURIPS 2017). It’s unclear what the results in Figure 3 contribute in light of these baselines (besides the normalization by the constant C).

Which loss function was used to produce the results in Table 1? If it’s the loss in (25), it would also be useful to see calibration metrics for NLL loss alone.

Figure 2 shows strong sensitivity of ACE to the number of bins. Quantile ECE (an ECE metric with bins defined by quantiles instead of fixed-width) often shows less sensitivity -- was this metric considered as well?

---

> ### Author Response · Authors · 2020-11-12
> **Regarding novelty**
>
> Dear AnonReviewer2, thank you very much for your valuable feedback. In the following, we try to address every raised concern and hope to meet your expectations.
>
> Thank you for pointing out two relevant papers, one of which we have already taken into account. We will review the other and consider it in our manuscript. Your main concern seems to be the lack of novelty of our contribution. We do not propose the use of (normalized) entropy for measuring uncertainty as sole contribution, since this has been extensively studied in the papers you mentioned. Rather, the proposed novelty is a metric for measuring calibration (of a classification model) based on normalized entropy. Recent well-recognized papers have highlighted the lack of a suitable calibration metric and we aim to address this issue (Nixon et al., 2019; Ashukha et al., 2020; Kull et al., 2019; Kumar et al., 2019). The perfect calibration metric has yet to be found, and we believe to make a valuable contribution towards it. Our metric avoids pathologies of other metrics and has several favorable properties for measuring calibration (see p. 5). We hope that we have met your requirement for novelty.
>
> The contribution of Figure 3 is to show that for calibrated models, normalized entropy correlates with top-1 error, thus additionally providing an empirical justification for the use of normalized entropy in our metric (and definition of perfect calibration). The top-1 error decreases monotonically with the normalized entropy. Moreover, the results of Fig. 3 are novel, as Lakshminarayanan et al. (2017) only used max p for their rejection experiments. In our results, Bayesian methods perform much better as reported by Lakshminarayanan et al. (2017) and yield a top-1 error close to 0 for predictions with uncertainty < 0.1. We will add sentences to our manuscript for clarification.
>
> The results reported in Table 1 were produced using only NLL/cross-entropy loss. We will add the unregularized baseline to Table 2 for better comparison. Many thanks for this advice!
>
> We did not consider quantile ECE and will gladly include this in our revision. However, we did not find any related work describing quantile ECE in more detail. Can you point out a reference or describe quantile ECE briefly and how it differs from ACE?
>
> We hope that we have addressed all your concerns and are grateful if you update your rating of our work accordingly.
>
> References
>
> *see paper*

---

> > ### Comment · AnonReviewer2 · 2020-11-20
> > **Appreciate the clarifications**
> >
> > Thank you for clarifying the role of unnormalized entropy and the point you're making in Figure 3. Quantile ECE was used in Ovadia et al., 2019, "Can you trust your model's uncertainty? Evaluating predictive uncertainty under dataset shift", and reduces bias as compared with fixed-width ECE. Looking forward to your updated manuscript, after which I'll reevaluate my rating.

---

> > > ### Author Response · Authors · 2020-11-22
> > > **Regarding Quantile ECE**
> > >
> > > Please see Sect. 7 for our updated Table 2 as per your suggestion. We compare the regularization methods at optimal temperature as suggested by Ashukha et al. (2020).
> > > Thank you for pointing out this highly relevant paper, which we should have considered in the first place. We have carefully read the paper and did not find any comparison between quantile ECE and fixed-width ECE. After reviewing the provided source code (https://github.com/google-research/google-research/blob/master/uq_benchmark_2019/metrics_lib.py), we think that Ovadia et al. (2019) only used fixed-width bins in ECE computation. The only relevant mentioning seems to be:
> > >
> > > > When bins $ \\{\rho_s : s\in 1\ldots S \\} $ are quantiles of the held-out predicted probabilities, $|B_s|\approx|B_k|$ and the estimation error is approximately constant.
> > >
> > > We think that this describtion of using quantiles as bin edges is equivalent to ACE. Ovadia et al. (2019) state that the ECE estimation error is constant across all bins when using quantiles. However, we did not find any statement saying that quantile ECE is robust against a varying number of bins. Please let us know if we have misunderstood anything.

---

> > > > ### Comment · AnonReviewer2 · 2020-11-23
> > > > **quantile ECE**
> > > >
> > > > I believe you're correct that Ovadia et al. did not compare quantile and fixed-width ECE, and it is unclear in the paper and code which one they used (there is a `  get_quantile_bins`   method, but it doesn't appear to be called). Thank you for updating Table 2.
> > > >
> > > > In a response to another comment, you mentioned that you were running experiments on Fashion-MNIST and SVHN, are those results in?

---

> > > > > ### Author Response · Authors · 2020-11-24
> > > > > **Additional Experiments**
> > > > >
> > > > > We already added the results for SVHN and will add the results for Fashion-MNIST later today.

---

### Official Review · AnonReviewer4 · 2020-10-30

**Rating:** 4
**Confidence:** 4

**Review:**

Thanks for the interesting paper!

Summary

The authors focus on the important problem of improved calibration measures as compared to the (now fairly standard) expected calibration error (ECE). More specifically, they define a new "Uncertainty Calibration Error" (UCE) metric based on the normalized entropy of the predictive distribution, rather than the max probability (as in ECE). The metric still uses fixed-width binning (as in ECE), and they motivate the interpretation w.r.t. perfect calibration based on a theoretical limit. They provide a set of experiments to show the differences in model ranking, sensitivity to number of bins, etc. between UCE, ECE, etc. on various models.

Strengths

- As noted in previous literature (and referenced in this paper), improved measures of calibration is an important research area.
- The authors provide a great background section to place their research within the broader area of uncertainty research.
- The experiments on sensitivity to the number of bins is informative and highly relevant in the context of previous literature (e.g., Nixon et al. (2019)) where it has been shown that ECE is particularly sensitive to this setting.

Weaknesses

As I have noted in more detail down below, I believe this paper suffers from a few weaknesses. Overall, at the end of the paper as a reader, I'm still left with questions of whether UCE is truly a better calibration metric. As noted above, the insensitivity to number of bins is great and an improvement on ECE and ACE. However, I don't believe the remaining experiments make a strong case that the metric (1) provides a better measure of calibration, (2) yields consistently improved model performance when used as a regularizer (though it's interesting that it can be used as one!), or (3) allows for improved model selection. Additionally, I believe its interpretability is limited. As noted below, the experiments have mixed results or make comparison claims that detract from the overall message, which I find troubling from an experimental rigor standpoint. Furthermore, I think this paper would benefit from experiments that are set up such that they can directly measure and compare the ability of the metrics to measure calibration error.

Recommendation

Given the above strengths and weaknesses, I'm currently inclined to suggest rejection of the paper in its current form. However, I think this could be a great paper and as a community I don't believe we have yet to devise the perfect calibration metric -- perhaps this could be it! I would highly recommend the authors push on the points above.

Additional comments

- p. 3, 4: It could be informative to includes notes about NLL and Brier score being strictly proper scoring rules (Gneiting & Raftery, 2007; Parmigiani & Inoue, 2009) that theoretically should be maximized only when the forecaster emits the distribution they believe to be true, and thus should, in theory, be well-calibrated asymptotically. However, we indeed know from Guo et al. (2017) that empirically, models can still overfit, leading to poor calibration.
- p. 4: The definition of perfect calibration can be traced back to Brier (1950), and, unlike ECE, is not limited to only the max predicted probability. Rather, for any predicted probability $p_k$ for class $k$, the probability that the true class is class $k$ should be equal to $p_k$ for all $p_k$ and all $k$. That is, $\mathbb{P}(Y = k | P_k = p_k) = p_k, \forall p_k \in [0, 1], \forall k \in [1, K]$.
- p. 5: UCE is based on an argument that normalized entropy approaches the top-1 error in the limit of number of class going to infinity. While this is interesting theoretically, this assumption seems too strong for empirical settings, and I think this affects the interpretability of the metric as claimed in the conclusion.
- p. 6, 7, Section 5.1: This section (and Figures 4 & 5 in the appendix) make claims that NLL and Brier score "fail at comparing calibration of models with different accuracy, as the metrics are always lower for models with better accuracy". I find this argument both surprising and confusing in terms of motivation. As strictly proper scoring rules, they should indeed have lower values for better probabilistic models. Although accuracy is a non-proper scoring rule, it should still correlate well with those strictly proper rules, so it is expected that the better models with lower NLL / Brier score will (typically with some variance) have higher accuracy (variance being due to the non-proper nature of accuracy). All strictly proper scoring rules can be decomposed into calibration and refinement terms (Parmigiani & Inoue, 2009; DeGroot & Fienberg, 1983), but in the non-decomposed setting, it is not expected that these rules would directly measure calibration. Therefore, given the focus on calibration measures, I'm confused as to the motivation behind comparing to NLL & Brier score directly (beyond the overconfidence analysis from Guo et al. (2017)) as a means of motivating the usefulness of UCE.
- p. 8, 12: In the regularization experiment, different regularization approaches are being compared in terms of calibration, but it's difficult to assess the results. In Table 2 (which needs an additional entry for the non-regularized result from Table 1), UCE regularization appears to improve accuracy, NLL, and Brier score over the non-regularized baseline. Interestingly though, NLL, Brier score, ECE, ACE, UCE, and MMCE (i.e, all metrics other than accuracy) point towards entropy-regularization being superior. It does result in a lower accuracy than UCE reg and the baseline, but by the other metrics (including the strictly proper scoring rules NLL and Brier score), it produces a better probabilistic model. For CIFAR-10 UCE reg is worse than the other regularization methods and the baseline.
- p. 8: Rejection & OOD Detection: This has been studied previously for unnormalized entropy, which should yield the same results. See, e.g., Malinin & Gales (2018), Ren et al. (2019).

Minor

- p. 3: s/as non Bayesian/as a non-Bayesian/
- p. 7: Figure 1 is too small.

---

> ### Author Response · Authors · 2020-11-17
> **Response**
>
> Dear AnonReviewer4, thank you for appreciating our work and for your thorough review. Below we try to respond to each issue raised and hope to meet your expectations.
>
> 1. Your main concern seems to be that we do not make a strong statement that our metric is beneficial. As it is difficult to highlight the distinct strengths of our metric in real-world experiments, we additionally conducted toy experiments that clearly show cases where UCE is able to capture miscalibration but other metrics fail. We emphasize that ECE and MMCE can be minimized by models with constant output and that ACE produces arbitrary values for a varying number of bins. We are confident that our metric provides reasonable benefits and can be useful to the community. We will highlight the benefits more in our revision and hope that we can convince you of this as well.
> 2. When we compare at optimal temperature (as suggested by Ashukha et al. (2020)), UCE regularization is at least as good as MMCE reg. and outperforms entropy reg. (see Fig. 6 in appendix). We will add a table with metric values at optimal temperature to the main text to highlight the benefits of UCE regularization. However, using the UCE as a regularizer is not our main contribution and, as you have already mentioned, rather an interesting additional feature.
> 3. We think that improved model selection is mainly given by avoiding the pathologies of the other metrics. Secondly, we argue that UCE is as interpretable as ECE and easier to understand for practitioners than e.g. MMCE. We would gladly conduct another experiment that can directly measure and compare the metrics, if you have any suggestion.
>
> Answers to additional comments
>
> 1. Thank you for your suggestion. We will include notes about NLL and Brier being strictly proper scoring rules.
> 2. Thank you for pointing out that the definition of perfect calibration can be traced back beyond Guo et al., (2017). We will revisit Brier (1950) and update our manuscript accordingly.
> 3. Thank you for this comment. We agree that our assumption is strong for small numbers of classes (e.g. $C < 10$). However, we think that this assumption is reasonable in empirical settings, where $ C = 100 $ (CIFAR-10) or $ C = 1000 $ (ImageNet). We explicitly mention the multi-class setting for our metric in the title of our paper. To shed additional light on this, we will add a figure to our manuscript that shows the effect of an increasing number of classes on the normalized entropy and will discuss this caveat in our conclusion.
> 4. Thank you for pointing out this confusion. As you suggested above, we will add a note about strictly proper scoring rules to our revision. Further, we will visually separate the NLL and Brier score values in Table 1 in order to not directly compare calibration metrics to strictly proper scoring rules.
> 5. As already mentioned above, UCE regularization outperforms entropy regularization when compared at optimal temperature (Ashukha et al., 2020). This can already be seen in Figures 6 & 7 and holds for both CIFAR-10 and CIFAR-100. We will further highlight this in our revision. Additionally, we will follow your suggestion and add the non-regularized baseline to the table.
> 6. Thank you for pointing out relevant prior work. We will consider this on our revision. It is correct that the rejection experiments have already been conducted with unnormalized entropy. Our experiments aim at highlighting the interpretability of UCE/normalized entropy: Rejecting test samples where $ \tilde{\mathcal{H}} > 0.2 $ will result in a classification error $< 0.2$ for the remaining samples if the model is well-calibrated (see Fig. 9 & 2). We argue that using normalized entropy as uncertainty measure is as interpretable as max p, but avoids the pathologies of max p when used in a calibration metric. We will make this clearer in the text.
>
> Thank you again for your detailed review. We hope to have taken all concerns into account and are currently working hard on the revision of our manuscript. We hope that we can convince you of our work and acknowledge an update of your rating. We are open for any further discussion.
>
> References
>
> *see paper*

---

### Author Response · Authors · 2020-11-12
**General Response**

We thank all reviewers for their valuable feedback, as it helps us to improve our paper considerably. We are currently conducting additional experiments as requested by the reviewers and will update the manuscript accordingly. We welcome an open discussion and are working hard to address all issues raised.

---

### Public Comment · ~Jize_Zhang1 · 2020-11-13
**Related work using KDE to mitigate the bias/binning issues in calibration error estimation**

Please check out our recent work on the use of KDE-based ECE estimator [1]. By replacing histogram with KDE, we provide a more reliable evaluation of the calibration error while mitigating the bias & binning sensitivity of existing histogram ECE estimators. The code is also available online.

[1] Jize Zhang, Bhavya Kailkhura, and T Han. "Mix-n-Match: Ensemble and compositional methods for uncertainty calibration in deep learning.", ICML 2020, https://arxiv.org/pdf/2003.07329.pdf

---

> ### Author Response · Authors · 2020-11-13
> **Relevant Work**
>
> Dear Jize Zhang, thank you for pointing out your highly relevant work. We will review your paper and consider it in our upcoming manuscript.

---

### Author Response · Authors · 2020-11-24
**Please See Rebuttal Version**

Dear reviewers, we thank you again for your valuable feedback that helps us greatly to improve our manuscript. Please see § 7 in the rebuttal version for new and updated results. We will integrate the parts of § 7 into the main text and update the rest of the manuscript according to your suggestions in a possible camera-ready version. We hope to have addressed all raised concerns and appreciate an update on the rating.

---

### Decision · Program_Chairs · 2021-01-07
**Final Decision**

**Decision:**

Reject

**Comment:**

This work proposes a novel metric for measuring calibration error in classification models.

Pros:
* Novel calibration metric addressing limitations of previously used metrics such as ECE

Cons:
* Limited experimental validation on CIFAR-10/CIFAR-100 only
* Unclear impact beyond proposing a new calibration metric
* Unclear value of using the proposed UCE metric for regularization and OOD detection

All reviewers appreciate the aim of the work to produce a calibration metric that addresses shortcomings of commonly used existing metrics such as expected calibration error (ECE), which is known to be sensitive to discretization choices.  However, all reviewers remain in doubt whether the proposed metric (uncertainty calibration error, UCE) is truly a better metric of calibration than previous proposals.  This doubt comes from two sources: 1. limited experiments that do not convincingly show the usefulness of UCE; and 2. interpretability of UCE not being as intuitive to the reviewers.  The experiments also use UCE as regularizer but the benefit of doing so over simple entropy regularization is not clear.

Overall the work is well-motivated and written and the proposed UCE measure is interesting.  However, the reviewers remain unconvinced of the claimed benefits and the potential impact for measuring or improving calibration.